# Induced Resistance Combined with RNA Interference Attenuates the Counteradaptation of the Western Flower Thrips

**DOI:** 10.3390/ijms231810886

**Published:** 2022-09-17

**Authors:** Tao Zhang, Li Liu, Yulian Jia, Junrui Zhi, Wenbo Yue, Dingyin Li, Guang Zeng

**Affiliations:** Institute of Entomology, Guizhou University, Guizhou Provincial Key Laboratory for Agricultural Pest Management in the Mountainous Region, Guiyang 550025, China

**Keywords:** *Frankliniella occidentalis*, glutathione S-transferase, induced defense, jasmonic acid, methyl jasmonate, RNA interference, metabolites, counteradaptation

## Abstract

The western flower thrips, *Frankliniella occidentalis* Pergande, is an invasive pest that damages agricultural and horticultural crops. The induction of plant defenses and RNA interference (RNAi) technology are potent pest control strategies. This study investigated whether the anti-adaptive ability of *F*. *occidentalis* to jasmonic acid (JA)- and methyl jasmonate (MeJA)-induced defenses in kidney bean plants was attenuated after glutathione S-transferase (GST) gene knockdown. The expression of four GSTs in thrips fed JA- and MeJA-induced leaves was analyzed, and *FoGSTd1* and *FoGSTs1* were upregulated. Exogenous JA- and MeJA-induced defenses led to increases in defensive secondary metabolites (tannins, alkaloids, total phenols, flavonoids, and lignin) in leaves. Metabolome analysis indicated that the JA-induced treatment of leaves led to significant upregulation of defensive metabolites. The activity of GSTs increased in second-instar thrips larvae fed JA- and MeJA-induced leaves. Co-silencing with RNAi simultaneously knocked down *FoGSTd1* and *FoGSTs1* transcripts and GST activity, and the area damaged by second-instar larvae feeding on JA- and MeJA-induced leaves decreased by 62.22% and 55.24%, respectively. The pupation rate of second-instar larvae also decreased by 39.68% and 39.89%, respectively. Thus, RNAi downregulation of *FoGSTd1* and *FoGSTs1* reduced the anti-adaptive ability of *F*. *occidentalis* to JA- or MeJA-induced defenses in kidney bean plants.

## 1. Introduction

In the co-evolution of plants and insects, plants developed a series of constitutive and inducible defense mechanisms against insect feeding stress. However, insects also evolved complex anti-defense mechanisms, including behavioral, physiological, and biochemical adaptations, to increase survival and reproduction [1,2,3,4]. Important progress has been made in the theory of plant defense [5,6]. In defense against herbivorous insects, plants induce the production of secondary compounds in response to various induction factors, and those compounds cause direct toxicity to insects [7,8]. Phytohormones are important inducing factors that regulate plant defense responses, and predominantly include jasmonic acid (JA), salicylic acid, ethylene, and abscisic acid and its related derivatives [9]. Among the phytohormones, the JA signaling pathway is the most important pathway for regulating plant defenses against insects. Activation of the JA signaling pathway in plants activates corresponding defensive enzymes and proteins and a series of defensive compounds that counteract the harm caused by herbivorous insects [10,11,12]. To activate the plant JA defense system, the JA signaling pathway is induced by chewing and rasping/sucking by insect mouthparts and also by the exogenous synthesis of JA and methyl jasmonate (MeJA) [6,13]. The activation of the JA signaling defense network by exogenous JA and its derivatives significantly upregulates JA biosynthetic enzymes, such as lipoxygenase (LOX), antioxidant enzymes (such as polyphenol oxidase and peroxidase), and protease inhibitor activities. Exogenous JA and its derivatives can also induce secondary metabolite and volatile production. The resulting plant defense response affects insects in various ways, including reducing feeding and digestion, retarding growth and development, and inhibiting population reproduction [14,15,16,17]. For example, when peanut plants of different genotypes were treated with exogenous JA for 1 day, *Helicoverpa armigera* feeding was inhibited. The JA biosynthesis enzyme LOX and a series of antioxidant enzymes were significantly activated in peanut plants, and contents of secondary metabolites (total phenols, tannins, and flavonoids) increased, thereby increasing peanut resistance to *H*. *armigera* [11]. The exogenous administration of JA also increases resistance of tomato to *Spodoptera littoralis* by mediating biosynthesis of the alkaloid nicotine [18].

The adaptation of insects to plant defense responses is mainly manifested in the use of salivary proteases in the mouth, digestive proteases in the midgut, and detoxification metabolism-related enzymes, as well as in gene regulation to detoxify, metabolize, digest, and absorb plant defensive compounds [1,2,3,19]. Detoxification enzymes involved in regulating insect adaptation to host plant defense responses include cytochrome P450s (P450), esterases, and glutathione S-transferases (GSTs), all of which occur widely in insects and can be activated by plant secondary metabolites [2,15,20]. The GSTs are resistance-related primary and secondary metabolic enzymes in insects that are involved in the degradation of heterologous substances. Glutathione S-transferases also have vital roles in regulating insect metabolic detoxification of host plant defensive compounds [21,22,23]. Insect cytoplasmic GSTs comprise delta, epsilon, omega, sigma, theta, and zeta subfamilies [24]. However, metabolic detoxification of exogenous or endogenous toxic substances in insects does not require the participation of all subfamily members. Different insects can activate individual or several subfamily genes according to specific needs and to regulate metabolic detoxification of specific toxic compounds. For example, *Chironomus riparius* simultaneously activates genes in delta, sigma, and epsilon subclasses to adapt to oxidative stress [25]. *Drosophila melanogaster* activates sigma class *DmGSTS1-1* and delta class *DmGSTD1* to metabolize and detoxify plant 4-hydroxynonenal and mustard oils, respectively [26,27]. Similarly, *Nilaparvata lugens* mainly degrades the rice defensive toxin gramine by regulating the delta-class gene *NlGSTD2* and the epsilon-class gene *NlGSTE1*, which increases the adaptability of *N. lugens* to rice plants [28]. Although progress has been made in understanding how insect GSTs regulate the metabolism and detoxification of plant defensive compounds, the specific mechanism by which GSTs regulate insect adaptation to plant defense responses activated by exogenous inducers requires further clarification.

Western flower thrips, *Frankliniella occidentalis* Pergande, in the family Thripidae and order Thysanoptera, are a serious worldwide invasive pest. Because of their feeding, oviposition, and transmission of plant viruses, *F. occidentalis* is a major threat to the production and market value of important economic food crops and ornamental flowers [29,30]. Thrips are well concealed and have high fecundity, producing more than 15 generations per year on field crops, and long-term chemical control in the field has led to a significant increase in thrips’ resistance to pesticides [31,32]. Therefore, using plant-induced defense responses to increase crop resistance to *F*. *occidentalis* may provide a new strategy to reduce chemical pesticide applications to control *F*. *occidentalis* [33]. The use of induced defense responses to increase crop resistance to *F*. *occidentalis* has been investigated in many studies. Factors that induce defense responses include exogenous plant hormones and the derivatives, plant secondary metabolites, biological factors such as microorganisms, and abiotic factors such as ultraviolet (UV-B) rays and CO_2_ [13]. Among inducing factors, the exogenous plant hormone JA and its derivatives, such as MeJA and cis-jasmone, can induce increased resistance against *F*. *occidentalis* in *Arabidopsis* [34], Chinese cabbage, multiple genotypes of *Arabidopsis* [35], kidney bean [36,37], pepper [38], and chrysanthemum [39]. Although plant-induced resistance can reduce populations of *F. occidentalis* to a certain extent and reduce damage to crops, some *F. occidentalis* can adapt to resistant crops and survive and reproduce [40]. However, whether the *F. occidentalis* that survive long-term exposure to induced resistance in crops can adapt to the inducible defense response of plants by regulating detoxification enzymes and associated genes remains to be elucidated.

In summary, inducing plant defenses is an ecologically beneficial method of pest control. However, corresponding anti-adaptive abilities have also evolved in insects during long-term interactions of insects with plant defense responses [41]. The formation of such anti-adaptation mechanisms limits the application of plant-induced insect resistance. Therefore, identifying the molecular targets of insect adaptation to plant-induced resistance and then performing RNA interference (RNAi) to reduce insect anti-adaptation ability may improve pest control. The aim of the current study was to determine the specific GSTs of western flower thrips that actively respond to exogenous JA- and MeJA-induced kidney bean defense responses and then use RNAi technology to knock down the expression levels of those GST genes. The feeding adaptation of *F. occidentalis* to the defense response of kidney beans induced by exogenous JA and MeJA was investigated after knockdown of the key GST genes. This study provides a reference for related research on the combination of induced plant insect resistance and RNAi to reduce insect anti-adaptive ability and improve pest control effects.

## 2. Results

### 2.1. Response of Frankliniella occidentalis Glutathione S-Transferase Genes to Jasmonic Acid-Induced Defense Response in Kidney Bean

The expression of *FoGSTd1* and *FoGSTs1* was significantly upregulated in *F. occidentalis* second-instar larvae after feeding on exogenous JA-induced bean plants, whereas the expression of *FoGSTe1* and *FoGSTt1* was not significantly affected (Figure 1).

The expression of *FoGSTd1* was not significantly different from that in the control in *F. occidentalis* fed kidney bean leaves treated with 0.01 mmol/L exogenous JA for 1 d, but mRNA expression levels were significantly upregulated in all other treatments. When *F. occidentalis* was fed kidney bean leaves treated with 0.1 mmol/L JA for 1, 3, and 5 d, *FoGSTd1* expression was significantly upregulated 3.30-fold (*F*_2, 6_ = 28.596; *p* < 0.001), 1.25-fold (*F*_2, 6_ = 12.354; *p* = 0.007), and 1.94-fold (*F*_2, 6_ = 46.465; *p* < 0.001), respectively, compared with that in the control (0.0 mmol/L). When *F. occidentalis* was fed bean leaves treated with 0.01 mmol/L JA for 3 and 5 d, *FoGSTd1* expression increased significantly 1.36- and 2.78-fold, respectively, compared with that in the control.

The relative expression levels of *FoGSTs1* in *F. occidentalis* fed exogenous JA-induced kidney bean leaves were significantly upregulated in all treatments, except for leaves treated with 0.10 mmol/L JA for 3 d, which had expression levels that were not significantly different from those in the control. After *F. occidentalis* was fed leaves treated with 0.10 mmol/L JA for 1 and 5 d, *FoGSTs1* expression was significantly upregulated 2.97- and 1.60-fold, respectively, compared with that in the control. Furthermore, when *F. occidentalis* was fed kidney bean leaves treated with 0.01 mmol/L JA for 1, 3, and 5 d, the expression of *FoGSTs1* was significantly upregulated 2.52-fold (*F*_2, 6_ = 14.287; *p* = 0.005), 1.29-fold (*F*_2, 6_ = 10.763; *p* = 0.010), and 2.21-fold (*F*_2, 6_ = 44.520; *p* < 0.001), respectively, compared with that in the control.

### 2.2. Response of Glutathione S-Transferase Genes of Frankliniella occidentalis to Methyl Jasmonate-Induced Treatment of Kidney Bean

The expression levels of *FoGSTd1*, *FoGSTs1*, and *FoGSTe1* were different after *F. occidentalis* second-instar larvae were fed kidney bean leaves treated for 1, 3, and 5 d with MeJA (Figure 2a–c). However, the expression of *FoGSTt1* was not affected by exogenous MeJA-induced defense responses in kidney bean plants (Figure 2d). When second-instar larvae were fed kidney bean leaves treated for 1, 3, and 5 d with 1.5 mmol/L MeJA, the expression of *FoGSTd1* increased significantly 1.90-fold (*F*_2, 6_ = 20.567; *p* = 0.002), 1.60-fold (*F*_2, 6_ = 6.788; *p* = 0.029), and 1.44-fold (*F*_2, 6_ = 11.781; *p* = 0.008), respectively, compared with that in the control. Although the expression levels of *FoGSTd1* were all upregulated after 1.0 mmol/L MeJA-induced treatment, the upregulation of *FoGSTd1* was only significant at 1 d of induction treatment. After thrips fed on kidney bean leaves that were induced with 1.5 mmol/L MeJA for 1 and 3 d, *FoGSTs1* expression was significantly upregulated 2.24-fold (*F*_2, 6_ = 7.508; *p* = 0.023) and 1.41-fold (*F*_2, 6_ = 13.267; *p* = 0.006), respectively, compared with that in the control. With 1.0 mmol/L MeJA treatment for 1 and 3 d, *FoGSTs1* expression was upregulated 1.98- and 1.28-fold, respectively. The expression level of *FoGSTe1* was significantly downregulated in *F. occidentalis* fed kidney bean leaves treated for 1 d with 1.5 mmol/L or 1.0 mmol/L MeJA. By contrast, *FoGSTe1* expression was significantly upregulated in *F. occidentalis* fed kidney bean leaves treated with MeJA for 3 and 5 d. When *F. occidentalis* fed on kidney bean leaves induced with 1.5 mmol/L and 1.0 mmol/L MeJA for 5 d, the expression of *FoGSTe1* was significantly upregulated 1.35- and 1.61-fold (*F*_2, 6_ = 48.091; *p* < 0.001), respectively, compared with that in the control.

### 2.3. Jasmonic Acid and Methyl Jasmonate Induce Accumulation of Secondary Defensive Substances in Kidney Beans and Affect Activity of Glutathione S-Transferases of Frankliniella occidentalis

Spraying kidney bean leaves with JA (0.10 mmol/L) or MeJA (1.50 mmol/L) for 1 d significantly induced the accumulation of tannins, alkaloids, total phenols, flavonoids, and lignin in leaves. In addition, GST activity was significantly upregulated in *F. occidentalis* second-instar larvae fed treated kidney bean leaves (Figure 3). The determination of secondary biomass content showed that exogenous JA and MeJA markedly induced tannins, alkaloids, total phenols, flavonoids, and lignin in kidney bean leaves. Methyl jasmonate caused a more pronounced induction effect compared with that of JA, and contents of tannins, alkaloids, and lignin in kidney bean leaves treated with MeJA were significantly higher than those in both the control and JA-induced treatment groups. The tannin contents of kidney bean leaves in the MeJA-treated group were 1.36- and 2.02-fold higher than those in the control and JA-treated groups, respectively (*F*_2, 15_ = 172.615; *p* < 0.001), whereas the alkaloid contents were 1.36- and 2.02-fold higher than those in the control and JA treatment groups, respectively (*F*_2, 15_ = 55.183; *p* < 0.001). In addition, the lignin contents of kidney bean leaves in the MeJA-treated group were 1.80- and 2.15-fold higher than those in the control and JA treatment groups, respectively (*F*_2, 15_ = 117.569; *p* < 0.001). The activity of GSTs in second-instar larvae fed JA- and MeJA-treated bean leaves was upregulated 1.43- and 1.22-fold, respectively, compared with that in the control group (*F*_2, 12_ = 4.829; *p* = 0.029).

### 2.4. Analysis of Differential Metabolites in Kidney Bean Leaves Induced by Jasmonic Acid

The effects of JA-induced kidney bean leaf defensive metabolites on activity and gene mRNA levels of thrips GSTs were further characterized. An Ultra High-Performance Liquid Chromatography AB Triple TOF 6600 mass spectrometer system (UHPLC-Q-TOF MS) was used to detect differential metabolites in JA-induced and control kidney bean leaves. Orthogonal partial least-squares discriminant analysis (OPLS-DA) indicated that the JA and control groups were completely separated in the positive and negative ion modes, and metabolic profiles were different, with all samples within the 95% confidence interval. This result indicated that the metabolites detected in the JA and control groups were significantly different (Figure 4). To evaluate the model, parameters were obtained by seven-fold cross-validation, including R^2^Y > 0.99 and Q^2^ > 0.5 (Appendix A). Thus, the data model was stable and reliable.

The metabolites detected by UHPLCQ-TOF MS were identified by matching with the local database, Shanghai Applied Protein Technology. A total of 668 metabolites were identified in the positive ion mode, of which 25 were significantly different metabolites satisfying the criteria variable importance in the projection (VIP) > 1 and *p* < 0.05 (Appendix A). A total of 366 metabolites were identified in the negative ion mode, of which seven were significantly different metabolites satisfying the criteria VIP > 1 and *p* < 0.05 (Appendix A). In the positive ion mode, 21 differential metabolites were significantly upregulated in the JA group compared with the control group, whereas four were significantly downregulated (Figure 5a). In the negative ion mode, only one metabolite was significantly downregulated in the JA group compared with the control group, whereas six were significantly upregulated (Figure 5b). Significantly upregulated differential metabolites (10 with the highest fold change) in the JA group are listed in Table 1. The metabolites were members of different superclasses, with 12 lipids and lipid-like molecules, five organoheterocyclic compounds, five undefined compounds, two benzenoids, two alkaloids and derivatives, four organic acids and derivatives, one organic oxygen compound, and one phenylpropanoid and polyketide. Visual analysis of pairs of metabolite molecules with correlation coefficient |r| > 0.8 and *p* < 0.05 in the positive ion mode revealed the inhibitor bexarotene, the diterpenoid compound ent-17-hydroxy-16beta-kauran-19-al, and apo-12’-capsorubinal had the highest correlations in the correlation network (Figure 6a). Further enrichment analysis was performed using the KEGG (Kyoto Encyclopedia of Genes and Genomes, http://www.kegg.jp/, accessed on 20 August 2022) database (Figure 6b). The differential metabolites were significantly enriched in the metabolic pathway’s mineral absorption, protein digestion and absorption, and biosynthesis of amino acids. Three metabolites were enriched in the glucosinolate biosynthesis metabolic pathway, and two metabolites were enriched in the biosynthesis of various secondary metabolites metabolic pathway. The physicochemical properties of each differential metabolite were searched separately by PubChem (https://pubchem.ncbi.nlm.nih.gov/, accessed on 20 August 2022). Among the metabolites upregulated by JA, succinic acid, bexarotene, bufogenin, nicotinamide, and vinpocetine had biological toxicity and antioxidant effects.

### 2.5. Frankliniella occidentalis FoGSTs1 and FoGSTd1 Expression and Glutathione S-Transferase Activity Are Downregulated by RNA Interference

When ds*FoGSTd1*, ds*FoGSTs1*, and ds*FoGSTs1/d1* solutions were fed to second-instar larvae of *F. occidentalis*, mRNA expression levels of *FoGSTd1* and *FoGSTs1* were effectively downregulated (Figure 7). In addition, downregulation of *FoGSTd1* and *FoGSTs1* individually or simultaneously significantly downregulated GST activity (Figure 8). The silencing effect of ds*FoGSTd1* and ds*FoGSTs1/d1* solutions on the expression level of *FoGSTd1* led to significant downregulation by 61.22% and 61.02%, respectively, compared with expression in the control group fed ds*eGFP* (*F*_3, 8_ = 32.583; *p* < 0.001). The ds*FoGSTs1* and ds*FoGSTs1/d1* solutions significantly downregulated the expression level of *FoGSTs1* by 59.56% and 61.89%, respectively, compared with that in the control group fed ds*eGFP* (*F*_3, 8_ = 23.971; *p* < 0.001). The activity of GSTs in second-instar larvae treated with ds*FoGSTs1/d1* solution decreased significantly by 48.78%, 39.77%, and 23.28% compared with that in the groups fed honey solution (HS), ds*eGFP*, and ds*FoGSTs1* solutions, respectively, (*F*_4, 20_ = 52.012; *p* < 0.001). Although GST activity was significantly downregulated when fed ds*FoGSTd1* solution, it was not significantly different from that when fed ds*FoGSTs1* or ds*FoGSTs1/d1* solutions.

### 2.6. Co-Silencing of FoGSTd1 and FoGSTs1 Expression by RNA Interference Reduces the Feeding Ability of Frankliniella occidentalis

When the expression of *FoGSTd1* and *FoGSTs1* was co-silenced, the area of feeding damage by second-instar larvae of *F. occidentalis* on kidney bean leaves treated with exogenous JA and MeJA decreased significantly (Figure 9). The area of feeding damage on JA- and MeJA-induced kidney bean leaves by second-instar larvae fed HS decreased significantly by 28.82% and 38.49% (*F*_2, 15_ = 25.451; *p* < 0.001), respectively, compared with feeding on control leaves without induction treatment. Compared with the control, the area of feeding damage on JA- and MeJA-induced bean leaves by *F. occidentalis* fed ds*eGFP* solution decreased significantly by 26.35% and 30.31% (*F*_2, 15_ = 13.360; *p* < 0.001), respectively, whereas the area of feeding damage on JA- and MeJA-induced kidney bean leaves by *F. occidentalis* fed ds*FoGSTs1/d1* solution decreased significantly by 55.11% and 54.03% (*F*_2, 15_ = 48.189; *p* < 0.001), respectively.

Thrips were fed ds*FoGSTs1/d1* solution then fed kidney bean leaves treated with different induction factors (uninduced or induced by exogenous JA and MeJA) and feeding damage areas were measured. The area of feeding damage by *F. occidentalis* fed ds*FoGSTs1/d1* solution decreased significantly compared with that of both the HS and ds*eGFP*-treated groups. In the control group, the area of feeding damage on kidney bean leaves by *F. occidentalis* fed ds*FoGSTs1/d1* solution decreased significantly by 40.10% and 33.09% (*F*_2, 15_ = 22.344; *p* < 0.001), compared with *F. occidentalis* fed HS and ds*eGFP*, respectively. The damaged areas of the JA-induced treatment groups decreased significantly by 62.22% and 59.22%, respectively (*F*_2, 15_ = 30.856; *p* < 0.001), and those of the MeJA-induced treatment groups decreased by 55.24% and 55.87% (*F*_2, 15_ = 80.585; *p* < 0.001), respectively.

### 2.7. Co-Silencing of FoGSTd1 and FoGSTs1 by RNA Interference Reduces the Pupation Rate of Frankliniella occidentalis

The co-silencing of *FoGSTd1* and *FoGSTs1* expression levels significantly reduced the pupation rate of second-instar larvae of *F. occidentalis* fed exogenous JA- and MeJA-induced kidney bean leaves (Figure 10). In the group fed HS without dsRNA, the pupation rate of second-instar larvae of *F. occidentalis* feeding on exogenous JA- and MeJA-induced bean leaves decreased significantly by 21.25% and 26.98% (*F*_2, 15_ = 16.234; *p* < 0.001), respectively, compared with that in the control group. In the group fed ds*eGFP* solution, the pupation rate of second-instar larvae after feeding on JA- and MeJA-induced bean leaves decreased significantly by 14.29% and 15.75% (*F*_2, 15_ = 6.012; *p* = 0.012), respectively, compared with that feeding on the control. In the group fed ds*FoGSTs1/d1* solution, the pupation rate of second-instar larvae that fed on JA- and MeJA-induced bean leaves decreased significantly by 21.65% and 25.97% (*F*_2, 15_ = 8.994; *p* = 0.003), respectively, compared with that feeding on the control.

The second-instar larvae were pre-fed ds*FoGSTs1/d1* solution and then transferred to leaves of control, JA-, and MeJA-induced kidney bean leaves for rearing. Pupation rates were significantly lower than those of larvae pre-fed HS and dse*GFP* solution. In the control group, the second-instar larvae fed ds*FoGSTs1/d1* had significantly lower pupation rates by 39.38% and 34.01% (*F*_2, 15_ = 49.674; *p* < 0.001) than those of larvae fed HS and ds*eGFP* solution, respectively. In the JA group, the pupation rate decreased significantly by 39.68% and 39.68% (*F*_2, 15_ = 41.126; *p* < 0.001), and in the MeJA group, the pupation rate decreased significantly by 38.89% and 42.78%, respectively (*F*_2, 15_ = 40.583; *p* < 0.001).

## 3. Discussion

Because the injury-related plant hormones and signaling molecules JA and MeJA are widely distributed in plants, exogenous application can effectively trigger plant physiological and biochemical defense systems to increase resistance to herbivorous insects [12,42,43]. Insects have evolved mechanisms to adapt to various defense responses of host plants, especially toxic secondary metabolites produced by plants. Insects can use detoxification enzyme systems such as cytochrome P450 enzymes, carboxylesterases, and GSTs in the gut and fat body to weaken the activity of anti-insect substances [2,19]. Among detoxification enzymes, cytochrome P450 is the best-studied enzyme system for detoxification and metabolism of exogenous or endogenous toxic compounds in insects [44]. By contrast, the current study focused predominantly on GSTs. The expression levels of *FoGSTd1* and *FoGSTs1* mRNA were significantly upregulated after second-instar larvae of *F. occidentalis* fed on kidney bean leaves treated for 1, 3, and 5 d with different concentrations of JA and MeJA. However, *FoGSTe1* and *FoGSTt1* did not respond positively to JA- and MeJA-induced kidney bean defense processes.

Results in the current study are consistent with the conclusions of previous studies that not all GST family members are involved in regulating insect metabolic detoxification of toxic compounds at the same time [28,45]. For example, Huang et al. [46] identified eight GST genes from *Spodoptera litura*, but only *SlGSTe1*, *SlGSTe3*, *SlGSTs1*, *SlGSTs3*, and *SlGSTo1* were highly expressed in response to xanthotoxin stress. Further research found that *S. litura SlGSTE1* has a key role in detoxifying glucosinolates and furocoumarins [47]. The seven GST genes of *N. lugens* are highly expressed in response to the rice defensive toxin gramine, but the crucial genes for the detoxification of gramine are *NlGSTD2* and *NlGSTE1* [28]. In addition, the detoxification of specific toxic compounds by genes of the same subfamily also requires the activation of different genes. For example, *AgosGSTs1* of the *Aphis gossypii* Sigma subfamily binds to the insecticide piperonyl butoxide, whereas *AgosGSTs2* binds to the natural compound tannin of plant origin [48]. Similar findings are reported with cytochrome P450 superfamily enzymes. For example, among the six P450 genes of *H. armigera*, *CYP4L11*, *CYP6AB9*, and *CCE001b* are instrumental in the detoxification and metabolism of gossypol [49], whereas *S. litura* may utilize only a single detoxification gene, *CYP6AB60*, to detoxify toxic secondary metabolites in different host plants [50].

Exogenous JA and MeJA can induce plants to activate the JA defense network, and secondary metabolites in the JA defense network are compounds that can be directly toxic to insects [11,15,16,17]. In the present study, whether the significantly upregulated expression of *FoGSTd1* and *FoGSTs1* in *F. occidentalis* was related to the accumulation of plant defensive secondary metabolites induced by exogenous JA and MeJA was investigated. Thus, contents of secondary metabolites (tannins, flavonoids, total phenols, alkaloids, and lignin) in kidney bean leaves were determined. JA and MeJA significantly induced the accumulation of those five secondary metabolites in kidney bean leaves. When War et al. [11] applied JA to peanuts, the accumulation of secondary metabolites such as peanut tannins, flavonoids, and total phenols was also induced, which subsequently increased the resistance of peanuts to *H. armigera*. However, insects detoxify and metabolize plant defensive secondary metabolites by regulating detoxification enzymes [2]. For example, GSTs of *Myzus persicae* metabolize isothiocyanates in crucifers [51], and phenolic glycosides increase GST activity in *Helopeltis theivora* [52]. In the current study, GST activity increased after *F. occidentalis* ingested JA- and MeJA-induced kidney bean leaves, with GST activity increasing most significantly in the JA-induced treatment group. The secondary biomass content of MeJA-induced kidney bean leaves was higher than that of the JA-induced treatment, but the GST activity of *F. occidentalis* fed MeJA-induced kidney bean leaves was lower than that of the JA-treated group. Those results might have occurred because more thrips GST enzymes were consumed in detoxifying and metabolizing MeJA-induced bean secondary metabolites.

Metabolites are at the core of interactions between cell changes and phenotypes, which directly reflect the physiological state of cells [53]. Plant hormones can drive plant defense signaling pathways to promote up- or downregulation of plant endogenous metabolites to accurately indicate plant defense phenotypes against herbivores. Primary metabolites are precursors for the synthesis of secondary metabolites, which are important in plant defense against herbivores [54]. Previous studies show that JA and MeJA can induce upregulation or downregulation of plant metabolites to increase plant defense against herbivorous insects [55,56]. Further analysis of specific metabolite differences in kidney bean leaves induced by JA and MeJA that led to the upregulation of thrips GST enzyme activity and gene mRNA levels was important in the in-depth analysis of the defense responses induced by JA and MeJA in kidney bean plants. Both JA and MeJA are produced through the octadecanoic acid pathway. When exogenous MeJA is applied on plants, it activates the JA signaling pathway through conversion to JA and JA-isoleucine [57,58]. Therefore, in this study, metabolite differences based on UHPLC-Q-TOF MS were only detected between the JA and control groups. The results showed that compared with the control group, JA treatment induced a significant upregulation of 27 metabolites and a significant downregulation of 5 metabolites in kidney bean leaves. Among the significantly upregulated metabolites, succinic acid, bexarotene, bufogenin, nicotinamide, and vinpocetine had a certain degree of biological toxicity and antioxidant effects. JA is derived from lipids [59] and 12 of the 32 differential metabolites detected in this study were lipids and lipid−like molecules. Compared with the control group, 10 lipids and lipid-like molecules were significantly upregulated in JA-treated kidney bean leaves. In addition, tryptophan, l-isoleucine, and the JA derivative jasmolone, which are closely related to JA biosynthesis, were also significantly upregulated. However, the jasmolone analog cis-jasmone can inhibit thrips feeding damage to kidney bean leaves [36]. The results indicated that the enzyme activities and mRNA expression levels of thrips GSTs after feeding on JA and MeJA-induced bean leaves might be related to changes in the levels of those differential metabolites.

Exogenous JA- and MeJA-induced treatment of kidney bean resulted in a decrease in the feeding ability and pupation rate of second-instar larvae of *F. occidentalis* in the current study. This result is similar to results in other crops, in which the effect of JA and its derivatives is to induce an increase in resistance to *F. occidentalis*. For example, spraying tomato plants with JA resulted in a 75% reduction in the number of *F. occidentalis* compared with that in controls [34]. Jasmonic acid-induced treatment of *Arabidopsis thaliana*, Chinese cabbage, and pepper resulted in decreased feeding and spawning preferences of *F. occidentalis* [35,38,60]. Moreover, the mixed treatment of MeJA, cis-jasmone, and allyl anisole had a greater inhibitory effect on the feeding and spawning of *F. occidentalis* than that of the control [37,40]. Such results may be related to JA- and MeJA-induced increases in contents of plant secondary metabolites that cause direct toxicity to *F. occidentalis* or inhibit digestion and metabolism. Phenolics and alkaloids can reduce the feeding preference of *F. occidentalis* for kidney bean, resulting in reduced survival [61,62].

This study also used RNAi co-silencing technology to knock down the mRNA expression levels of *FoGSTd1* and *FoGSTs1* in second-instar larvae of *F. occidentalis*. The knockdown resulted in a decrease in GST activity. After feeding simultaneously on two target gene dsRNA solutions, the area of feeding damage by thrips on JA- and MeJA-induced kidney bean leaves decreased significantly, accompanied by a significant reduction in the pupation rate of second-instar nymphs. This result indicated that *FoGSTd1* and *FoGSTs1* positively regulated the adaptation of *F. occidentalis* to JA- and MeJA-induced defense responses in kidney bean. Studies in other insects also demonstrate that RNAi downregulates the expression levels of GST genes, which can lead to reduced adaptive capacity in insects. For example, RNAi knockdown of the *Tribolium castaneum* Sigma class GST gene (*GSTS6*) can lead to a decrease in antioxidant enzyme activity and weakened antioxidant capacity [63]. Furthermore, downregulation of *NlGSTD2* and *NlGSTE1* expression levels by RNAi resulted in reduced adaptation of *N. lugens* to the rice defensive toxin gramine [28]. Moreover, downregulation of *SlGSTE1* expression decreased the ability of *S. litura* to detoxify glucosinolates and furanocoumarins, thereby increasing host resistance to *S. litura* [47].

In conclusion, the genes *FoGSTd1* and *FoGSTs1* of second-instar nymphs of *F. occidentalis* actively responded to exogenous JA- and MeJA-induced kidney bean defense responses. Exogenous JA and MeJA increased the contents of tannins, flavonoids, total phenols, alkaloids, and lignin in kidney bean leaves. In addition, UHPLC-Q-TOF MS detection showed that exogenous JA induced a significant upregulation of endogenous defensive metabolites in kidney bean leaves. Those results led to an increase in GST activity in the second-instar larvae of *F. occidentalis* that fed on that treatment. Further downregulation of *FoGSTd1* and *FoGSTs1* mRNA expression and GST activity by RNAi co-silencing technology significantly decreased the feeding ability and pupation rate of second-instar larvae that fed on JA- and MeJA-induced kidney bean leaves. This study provides a reference for future studies inducing defense responses to increase crop resistance and demonstrates that RNAi can reduce the detoxification metabolism of insects to improve pest control effects.

## 4. Materials and Methods

### 4.1. Insect and Plant Cultivation

Western flower thrips, *Frankliniella occidentalis*, were collected from Huaxi District (26.43° N, 106.67° E), Guiyang, China. Thrips were reared in an RXZ multi-stage programming artificial climate box (Ningbo Jiangnan Instrument Factory, Ningbo, China) with kidney bean pods as the food source. Second-instar larvae reared for more than 60 generations were used as experimental insects. Thrips were reared at an average temperature of 25 ± 1 °C, relative humidity (RH) of 70% ± 5%, and a photoperiod of 14 h.

Kidney bean (*Phaseolus vulgaris* ‘Jinshulu’) seeds were obtained from the Shengnong Seed Company in Xinji City, Hebei Province, China. Seeds were planted in sterile nutrient soil (sterilized at 121 °C for 2 h) in an artificial climate room [25 ± 1 °C, 65% ± 10% RH, and 14:10 h (L:D) photoperiod]. One seed was planted in each vegetative pot (11.5-cm diameter, 10.0-cm height). Plants were kept free of pests and diseases during the growth period. Plants that were approximately the same size when the first trifoliate leaf emerged or approximately 12 d post-germination were used in all experiments.

### 4.2. Responses of Glutathione S-Transferase Genes of Frankliniella occidentalis to Jasmonic Acid- and Methyl Jasmonate-Induced Defense Responses in Kidney Bean

Jasmonic acid (0.01 and 0.10 mmol/L) and MeJA (1.0 and 1.5 mmol/L) solutions were prepared [JA and MeJA were purchased from Sigma-Aldrich (Shanghai) Trading Co., Ltd., Shanghai, China], and deionized water served as the control. Each kidney bean leaf was sprayed with 5.0 mL of JA or MeJA solution at different concentrations for 1, 3, and 5 d (whole plants were sprayed wet). Second-instar larvae of *F. occidentalis*, which were starved for 6 h, were released on kidney bean leaves after spraying with JA and MeJA. Fifty *F. occidentalis* (25 thrips/leaf) were released on the first pair of true leaves of each kidney bean plant. Kidney bean plants with thrips were covered with clear plastic drums (25-cm diameter, 50-cm height) sealed with 200-mesh gauze at the top. After 24 h of feeding by *F. occidentalis*, 30–35 second-instar larvae were collected from each kidney bean plant as an experimental sample. The experiment was set up with three biological replicates per treatment, with one kidney bean plant per biological replicate.

Total RNA of the experimental samples was extracted using the Eastep® Super Total RNA Extraction Kit (Promega (Beijing) Biotech Co., Ltd., Beijing, China). The total RNA of each sample was used as a template, and cDNA was synthesized by reverse transcription using the RevertAid First Strand cDNA Synthesis Kit (Thermo Scientific, Vilnius, Lithuania). *Frankliniella occidentalis EF-1a* (GenBank accession no. AB277244.1) was used as the internal reference gene [64]. The CFX96TM real-time fluorescent quantitative PCR system (BioRad, Hercules, CA, USA) and FastStart Essential DNA Green Master kit (Roche Diagnostics, Penzberg, Germany) were used to quantify *F. occidentalis FoGSTd1* (GenBank accession no. MT454918), *FoGSTs1* (GenBank accession no. MT454919), *FoGSTt1* (GenBank accession no. MT454920), and *FoGSTe1* (GenBank accession no. MT454921) mRNA transcript levels. The primers used for each gene in reverse-transcription quantitative PCR (RT-qPCR) are shown in Appendix A. The annealing temperature was Tm = 57 °C.

### 4.3. Determination of Secondary Metabolites in Kidney Bean Leaves

One day after kidney bean plants were sprayed with 0.10 mmol/L JA, 1.50 mmol/L MeJA, or deionized water, 1.0 g was sampled from the first pair of true leaves of each kidney bean plant in each treatment to determine contents of tannins, alkaloids, total phenols, flavonoids, and lignin. The experiment had six biological replicates per treatment and one bean plant per replicate.

Contents of tannins, total phenols, and flavonoids were determined by the sodium tungstate–phosphomolybdic acid colorimetric method, folin phenol method, and sodium nitrite–aluminum nitrate colorimetric method, respectively [65]. Lignin content was determined by the acetylation method [66], whereas alkaloid content was determined by the bromocresol green indicator method [67].

### 4.4. Detection of Metabolites in Kidney Bean Leaves by UHPLC-Q-TOF MS

One day after spraying kidney bean plants with 0.10 mmol/L JA or deionized water, 1.0 g was sampled from the first pair of true leaves of each kidney bean plant in each treatment for metabolome analysis of kidney bean leaves. The experiment had five biological replicates per treatment and one bean plant per replicate. Differential metabolites were detected using an Ultra High Performance Liquid Chromatography AB Triple TOF 6600 mass spectrometer system (UHPLC-Q-TOF MS) (AB SCIEX, Foster City, CA, USA) [68].

### 4.5. Determination of Glutathione S-Transferase Activity in Frankliniella occidentalis

One day after kidney bean plants were sprayed with 0.10 mmol/L JA, 1.50 mmol/L MeJA, or deionized water, 60 second-instar thrips larvae starved for 6 h were released on each bean leaf. After thrips fed for 24 h, 40–45 thrips per sample were collected to determine activity of GSTs. The experiment had five biological replicates per treatment. Activity of GSTs was determined according to the method of Zhi et al. [69] and using a GST assay kit and Coomassie brilliant blue (G-250), both purchased from Suzhou Keming Biotechnology Co., Ltd., Suzhou, China. The determining principle is that GST catalyzes the combination of glutathione and 1-chloro-2,4-dinitrobenzene to produce a product with a characteristic absorption peak at 340 nm.

### 4.6. FoGSTs1 and FoGSTd1 RNA Interference

Primers for dsRNA synthesis of *FoGSTs1* and *FoGSTd1* were designed using the online tool https://www.flyrnai.org/cgi-bin/RNAi_find_primers.pl (accessed on 10 May 2021) (Appendix A). Experiments related to dsRNA synthesis and RNAi were performed according to the methods of Zhang et al. [70]. Honey water (10%) filtered through a 0.22-µm bacterial filter was mixed with ds*FoGSTs1*, ds*FoGSTd1*, and ds*FoGSTs1/d1* (mixing of ds*FoGSTs1* and ds*FoGSTd1* dsRNA solutions) to prepare dsRNA solutions of 300 ng/µL. Following the experimental setup of Zhang et al. [70], 40 second-instar thrips larvae starved for 6 h were fed dsRNA solution. After 24 h, 30 surviving thrips were collected to extract total RNA. The silencing effect of feeding dsRNA on *FoGSTs1* and *FoGSTd1* mRNA expression levels was determined by RT-qPCR. The experiment included two control treatments: thrips fed HS without dsRNA and thrips fed HS with ds*eGFP.* Each treatment had three biological replicates. In addition, the same treatment samples were collected to determine GST activity, with five biological replicates per treatment.

### 4.7. Feeding Damage Area of Second-Instar Larvae of Frankliniella occidentalis on Kidney Bean Leaves

One day after kidney bean plants were sprayed with 0.10 mmol/L JA, 1.50 mmol/L MeJA, or deionized water (control), one of the first pair of true leaves was placed in a Petri dish (15-cm diameter). Subsequently, second-instar larvae, after feeding on 10% HS, ds*eGFP*, or ds*FoGSTs1/d1* solution (300 ng/µL) for 24 h, were released on kidney bean leaves placed in Petri dishes. The *F. occidentalis* were enclosed within a double-pass glass tube (3.5-cm diameter) closed at one end with a 200-mesh gauze. The junction between the glass tube and the bean leaf was surrounded with moist absorbent cotton to prevent *F. occidentalis* from escaping. After allowing *F. occidentalis* to feed for 72 h, the feeding damage area on the leaf was measured with transparent graph paper (1-mm diameter). The experiment included six biological replicates, with 30 *F. occidentalis* second-instar larvae in each biological replicate.

### 4.8. Pupation Rate of Frankliniella occidentalis

Second-instar larvae fed with dsRNA solution for 24 h were released on one leaf of the first pair of true leaves of kidney bean plants induced by JA and MeJA. The methods of dsRNA feeding and JA/MeJA induction treatment were the same as in Section 2.6. Kidney bean leaves were placed in a special Petri dish, as described in Section 2.6, and covered with a special Petri dish lid. Parafilm was used to seal the Petri dish to prevent *F. occidentalis* from escaping. The specially made lid had a square hole with a side length of 8 cm, which was then sealed with a 200-mesh gauze. The lid ensured breathability while also preventing *F. occidentalis* from escaping. After 6 d of treatment, the surviving pupae in each treatment were enumerated to calculate the pupation rate. In the experiment, six biological replicates were set for each treatment, and each replicate included 30 second-instar larvae of *F. occidentalis*.

### 4.9. Statistical Analyses

Relative gene expression was calculated using the 2^−^^△△Ct^ quantitative method [71]. The software package SPSS v21.0 (IBM, Armonk, NY, USA) was used for one-way ANOVA, and Tukey’s method was used for multiple comparisons (α = 0.05). SigmaPlot v14.0 (Systat Software, Inc., San Jose, CA, USA) was used to construct statistical plots.

To analyze UHPLC-Q-TOF MS data, after being normalized to total peak intensity, processed data were analyzed by the R package (ropls) and subjected to multivariate data analyses, including Pareto-scaled principal component analysis (PCA) and orthogonal partial least-squares discriminant analysis (OPLS-DA). Seven-fold cross-validation and response permutation testing was used to evaluate the robustness of the model. The variable importance in the projection (VIP) value of each variable in the OPLS-DA model was calculated to indicate its contribution to the classification. Metabolites with VIP value > 1 were further tested with Student’s *t*-tests at univariate level to measure the significance of each metabolite, with *p*-values less than 0.05 considered statistically significant.

## Figures and Tables

**Figure 1 ijms-23-10886-f001:**
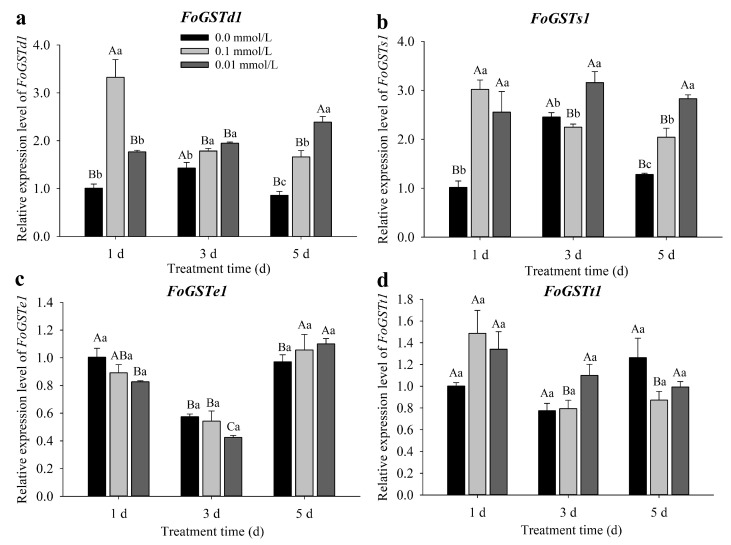
Relative expression levels of *FoGSTd1*, *FoGSTs1*, *FoGSTe1*, and *FoGSTt1* in second-instar larvae of *Frankliniella occidentalis* after feeding on exogenous jasmonic acid (JA)-induced kidney bean leaves. (**a**) Relative expression levels of *FoGSTd1*; (**b**) Relative expression levels of *FoGSTs1*; (**c**) Relative expression levels of *FoGSTe1*; (**d**) Relative expression levels of *FoGSTt1*. All data are the mean ± standard error (SE). Different uppercase letters above bars indicate significant differences in expression levels among different periods of the same treatment, whereas different lowercase letters indicate significant differences among different treatments at the same time (*p* < 0.05; one-way ANOVA, followed by Tukey’s test).

**Figure 2 ijms-23-10886-f002:**
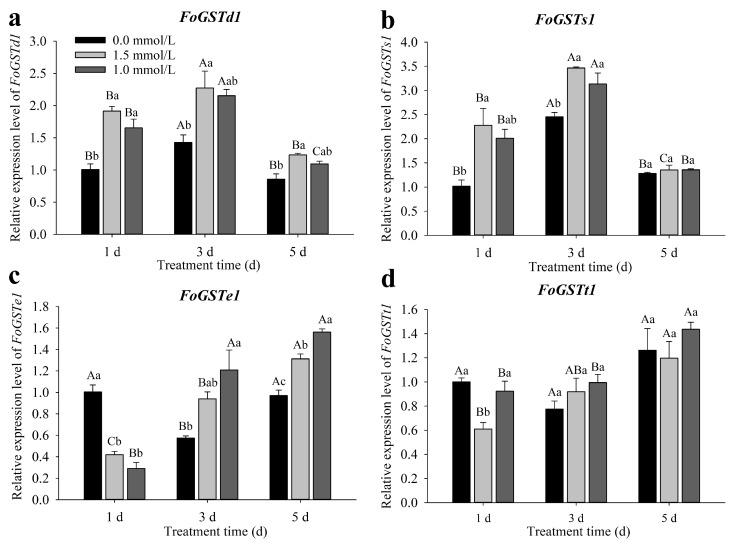
Relative expression levels of *FoGSTd1*, *FoGSTs1*, *FoGSTe1*, and *FoGSTt1* in second-instar larvae of *Frankliniella occidentalis* after feeding on exogenous methyl jasmonate (MeJA)-induced kidney bean leaves. (**a**) Relative expression levels of *FoGSTd1*; (**b**) Relative expression levels of *FoGSTs1*; (**c**) Relative expression levels of *FoGSTe1*; (**d**) Relative expression levels of *FoGSTt1*. All data are the mean ± SE. Different uppercase letters above bars indicate significant differences in expression levels among different periods of the same treatment, whereas different lowercase letters indicate significant differences among different treatments at the same time (*p* < 0.05; one-way ANOVA, followed by Tukey’s test).

**Figure 3 ijms-23-10886-f003:**
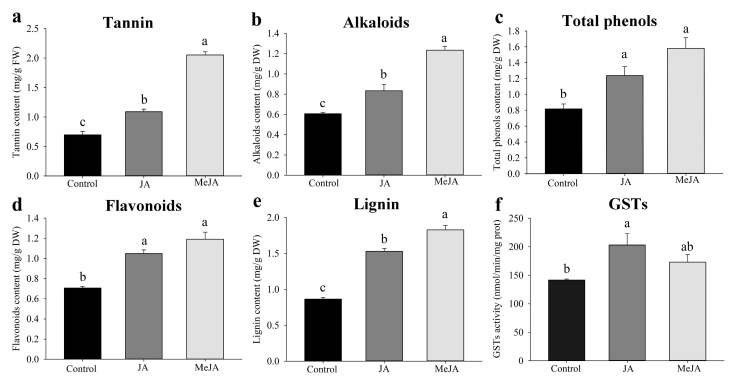
Jasmonic acid (JA) and methyl jasmonate (MeJA) induce secondary metabolite accumulation in kidney bean leaves and activate glutathione S-transferase (GST) activity in second-instar larvae of *Frankliniella occidentalis*. (**a**–**e**) are contents of tannins, alkaloids, total phenols, flavonoids, and lignin, respectively, in kidney bean leaves. (**f**) GST activity in second-instar larvae. Values are the mean ± SE. Different lowercase letters indicate significant deference among all treatments (*p* < 0.05; one-way ANOVA, followed by Tukey’s test).

**Figure 4 ijms-23-10886-f004:**
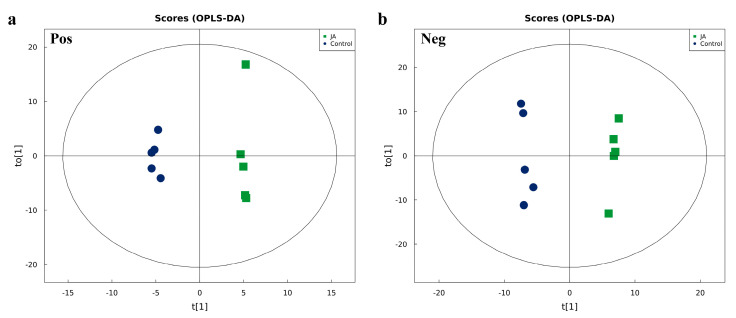
Orthogonal partial least squares discriminant analysis (OPLS-DA) score plot of jasmonic acid-treated and control samples. (**a**) Positive ion mode (Pos); (**b**) negative ion mode (Neg). The abscissa t[1] represents principal component 1, the ordinate t[1] represents principal component 2, and the ellipse represents the 95% confidence interval. Dots of the same color represent each biological replicate within a group, and dot distribution reflects the degree of difference between and within groups.

**Figure 5 ijms-23-10886-f005:**
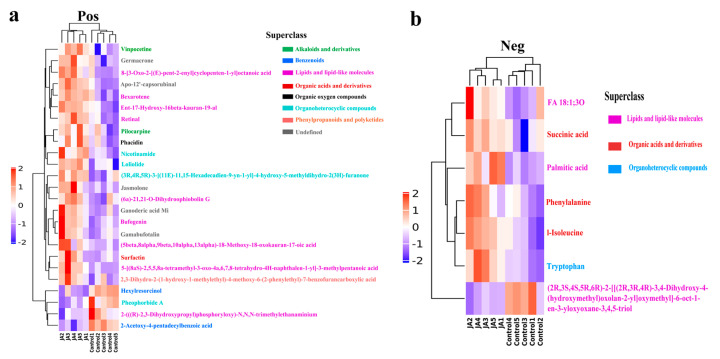
Heat map of differential metabolites in kidney bean leaves induced by jasmonic acid in positive (**a**) and negative (**b**) ion modes. Metabolite names of the same color are in the same superclass. Orthogonal partial least squares discriminant analysis (OPLS-DA) variable importance in the projection (VIP) > 1 and *p* < 0.05 were used as criteria to screen significantly different metabolites.

**Figure 6 ijms-23-10886-f006:**
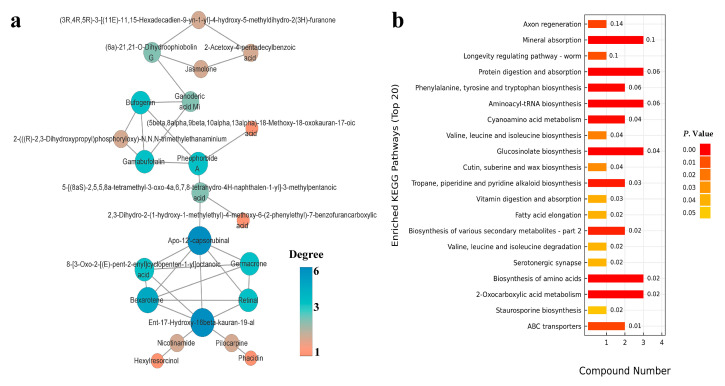
Differential metabolite correlation network in positive mode (**a**) and Kyoto Encyclopedia of Genes and Genomes (KEGG) pathway enrichment (**b**). The vertical axis in the bar graph represents each KEGG metabolic pathway, and the horizontal axis represents the number of differentially expressed metabolites contained in each KEGG metabolic pathway. The color represents the *p*-value of the enrichment analysis, and the darker the color is, the smaller the *p*-value and the more significant the degree of enrichment. The number on the column represents the enrichment factor, and the enrichment factor represents the ratio of the number of differential metabolites in the pathway to the number of annotated metabolites in the pathway.

**Figure 7 ijms-23-10886-f007:**
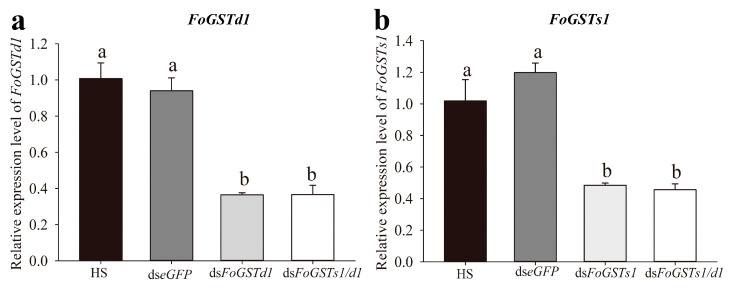
Downregulation of *FoGSTd1* and *FoGSTs1* expression levels in second-instar larvae of *Frankliniella occidentalis* when fed dsRNA solutions. (**a**) Relative expression level of *FoGSTd1*; (**b**) Relative expression level of *FoGSTs1*. Values are the mean ± SE. Different lowercase letters indicate significant silencing among all treatments (*p* < 0.05; one-way ANOVA, followed by Tukey’s test).

**Figure 8 ijms-23-10886-f008:**
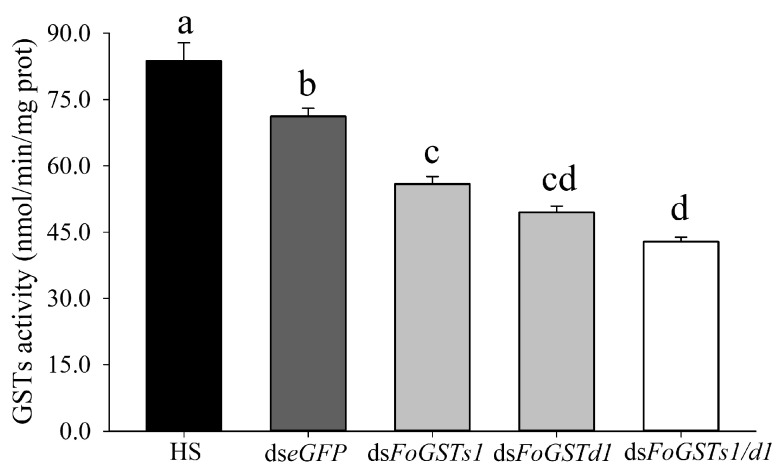
Effects of downregulation of *FoGSTs1* and *FoGSTd1* expression levels by RNA interference on glutathione S-transferase activity in second-instar larvae of *Frankliniella occidentalis*. Values are the mean ± SE. Different lowercase letters indicate significant silencing among all treatments (*p* < 0.05; one-way ANOVA, followed by Tukey’s test).

**Figure 9 ijms-23-10886-f009:**
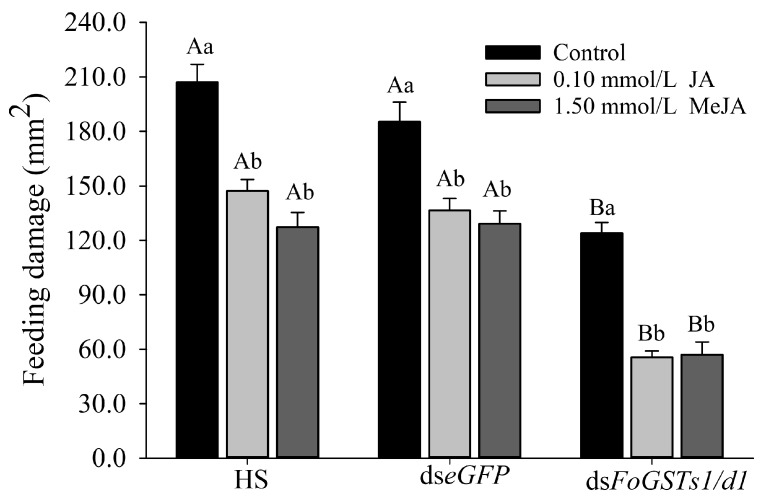
Co-silencing of *FoGSTd1* and *FoGSTs1* expression levels by RNA interference reduced the area of feeding damage on jasmonic acid (JA)- and methyl jasmonate (MeJA)-induced bean leaves by second-instar larvae of *Frankliniella occidentalis*. Values are the mean ± SE. Different lowercase letters indicate significant differences in feeding damage area on kidney bean leaves with different induction treatments by *F. occidentalis* fed the same RNAi solution. Different uppercase letters indicate significant differences in feeding damage area of kidney bean leaves with the same induction treatment, but *F. occidentalis* was fed honey solution (HS) and ds*eGFP* and ds*FoGSTs1/d1* solutions (*p* < 0.05; one-way ANOVA, followed by Tukey’s test).

**Figure 10 ijms-23-10886-f010:**
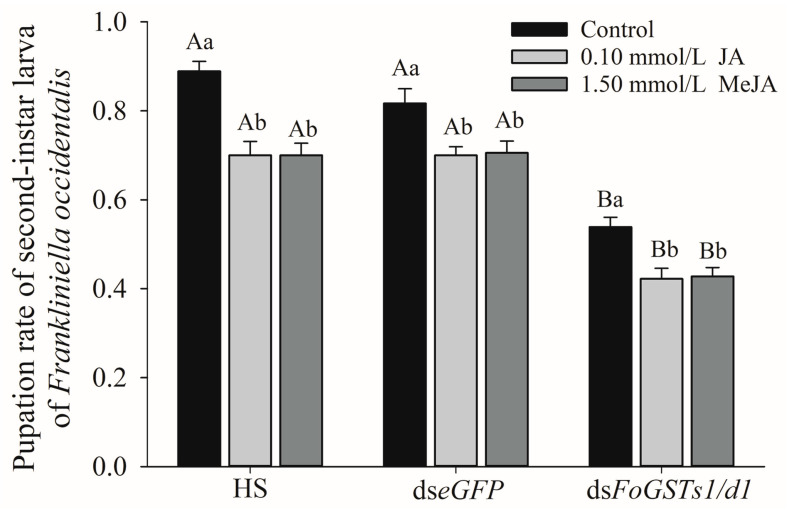
Co-silencing of *FoGSTd1* and *FoGSTs1* by RNA interference reduced the pupation rate of second-instar larvae of *Frankliniella occidentalis* feeding on jasmonic acid (JA)- and methyl jasmonate (MeJA)-induced kidney bean leaves. Values are the mean ± SE. Different lowercase letters indicate that the pupation rate of the second-instar larvae fed with the same solution was significantly different after feeding on the leaves of kidney bean with different induction treatments. Different uppercase letters indicate that the pupation rate of the second-instar larvae fed with different solutions was significantly different after feeding on the kidney bean leaves induced with the same treatment (*p* < 0.05; one-way ANOVA, followed Tukey’s test).

**Table 1 ijms-23-10886-t001:** Differential metabolites significantly upregulated in jasmonic acid-induced kidney bean leaves, compared with the control.

Name	Superclass	VIP	Fold Change	*p*-Value	Regulation
Palmitic acid (neg)	Lipids and lipid-like molecules	1.7415	2.0462	0.0015	up
Tryptophan (neg)	Organoheterocyclic compounds	1.2729	1.8842	0.0021	up
Phacidin (pos)	Organic oxygen compounds	1.2706	1.8176	0.0241	up
Nicotinamide (pos)	Organoheterocyclic compounds	1.8002	1.7992	0.0080	up
Succinic acid (neg)	Organic acids and derivatives	1.6529	1.4885	0.0366	up
Phenylalanine (neg)	Organic acids and derivatives	1.4458	1.4791	0.0445	up
Pilocarpine (pos)	Alkaloids and derivatives	1.1400	1.4764	0.0461	up
l-Isoleucine (neg)	Organic acids and derivatives	1.5679	1.4531	0.0037	up
Vinpocetine (pos)	Alkaloids and derivatives	3.6763	1.4100	0.0103	up
Jasmolone (pos)	Undefined	1.8603	1.1855	0.0195	up

Note: The differential metabolites upregulated in jasmonic acid treatment compared with the control group kidney bean leaves were the 10 with the highest fold change values. Variable importance for the projection (VIP): the larger the value, the more important it is. *p*-value indicates a significant difference. neg indicates the metabolite in negative ion mode. pos indicates the metabolite in positive ion mode.

## Data Availability

The data that support the findings of this study are available from the corresponding author upon reasonable request.

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
