# Peer review of "Induced Resistance Combined with RNA Interference Attenuates the Counteradaptation of the Western Flower Thrips"

_ijms, 2022, doi:10.3390/ijms231810886_

Round 1
Reviewer 1 Report (Previous Reviewer 1)
MDPI International Journal of Molecular Sciences
IJMS - 1926131
Induced Resistance Combined with RNA Interference Attenuates the Counteradaptation of Western Flower Thrips
The authors present a compelling, logical and sound body of research defining increased plant resistance to Thrips damage via inducing/enhancing plant defence responses and inducing knockdown of insect counters to plant defensive measures using RNAi. The authors show a logical progression of the research. Firstly, defining the insect GST subfamily members active in detoxification when fed leaves treated with JA or MeJA and the significant addition of analysis of the effects of applying JA or MeJA in the leaves. Secondly, focusing on the insect, the authors assessed the knockdown and activity of GST’ s using dsRNA molecules singularly and in combination, the effect of knockdown upon feeding area and pupation rate.
Accept.
Congratulations well done.
Reviewer 2 Report (Previous Reviewer 2)
Dear Authors,
I read the new version of the submitted manuscript IJMS-1926131 and I appreciated the great effort done to reach a high quality level of the text presentation . In particular, the high standard has been reached thanks to the improvement of the M&M and Results paragraphs . I also was satisfied to your replies to my notes at line 400 and 407 in the previous text. The short clarifications have been recorded at lines 498 and 503-505 of the new revised text. Thanks.

This manuscript is a resubmission of an earlier submission. The following is a list of the peer review reports and author responses from that submission.
Round 1
Reviewer 1 Report
Manuscript ID
ijms-1840176
Induced resistance combined with RNA interference attenuates the counteradaptation of Western flower thrips
Recommendation – Accept after minor revision
The authors present an excellent body of work with a logical approach. The authors demonstrated the exogenous application of JA or MeJA induced an increase in the defence response in plants. The regulation of GST family genes was assessed on induced plants to conclude that several of the GST family of genes were responsive in detoxifying the respective response induced in the plant. Further experimentation then focused on RNAi application to ‘knockdown’ the GST gene family members to define if disruption of the GST response in the context of a JA/MeJA inducted plants was able to affect feeding and pupation rate of F. occidentalis.
Syntax and English usage was appropriate.
Only a few comments to enhance readability
In the Abstract some idea of the significance of the reductions observed would be more exciting for the reader. Eg.
‘…feeding damage by…………leaves was reduced by x percentage’ and ‘The pupation rate……. was also reduced by ‘x’ percent.’
Consistency in the graphs. Eg between graph 5, 6 and 7 the fonts are larger in the axis’ of some and not others. Also, graphs titles would benefit ease of readability. Searching the y-axis to find out what was targeted was a pain. Put it in a title as the legend may not always be in the correct order of the graphs.
Other than those minor points an excellent piece of work – Congratulations.
Best regards.
Author Response
Response to Reviewer 1 Comments
Point 1: In the Abstract some idea of the significance of the reductions observed would be more exciting for the reader. Eg.
‘…feeding damage by…………leaves was reduced by x percentage’ and ‘The pupation rate……. was also reduced by ‘x’ percent.’
Response 1: We are grateful for the suggestion. Due to word limit, we only supplement the abstract with two representative sets of data. as follows:
RNAi co-silencing was used to simultaneously knockdown FoGSTd1 and FoGSTs1 transcripts and GSTs activity, and the area of feeding damage by F. occidentalis second-instar larvae on JA- and MeJA-induced kidney bean leaves was reduced by 62.22% and 55.24%. The pupation rate of the second-instar larvae was also reduced by 39.68 % and 39.89%.
Point 2: Consistency in the graphs. Eg between graph 5, 6 and 7 the fonts are larger in the axis’ of some and not others. Also, graphs titles would benefit ease of readability. Searching the y-axis to find out what was targeted was a pain. Put it in a title as the legend may not always be in the correct order of the graphs.
Response 2: Based on the reviewer's comments, we normalized the font size on the axes of Figures 5, 6, and 7 in the MS. In addition, we have added brief figure titles to each small figure in Figures 1, 2, 3, and 4 for readability. That is, each member of the composite graph is added with a label for easy reading. For revised figures, please see the attachment (manuscript).
Thank you again
Best regards.
All authors
Reviewer 2 Report
Dear Authors,
Please look at my short review of your manuscripts. It is the word attached file. There are only 2 "minor errors" I noted at line 400 and 407 of the Material & Methods section.
Line 400: The AA. have to record the geographic coordinates of the collection site. Also the original/natural host plant of WFT, has to be recorded.
Line 407: Why in a sterile soil? No basic nutrient element, ie. N, K, P, was in the raring soil? Please, explain briefly.

Author Response
Response to Reviewer 2 Comments
Point 1: Line 400: The AA. have to record the geographic coordinates of the collection site. Also the original/natural host plant of WFT, has to be recorded.
Response 1: We have added. as follows:
Western flower thrips, Frankliniella occidentalis, were collected from Huaxi District (Longitude and latitude: 106.67°E, 26.43°N), Guiyang, China.
Point 2: Line 407: Why in a sterile soil? No basic nutrient element, ie. N, K, P, was in the raring soil? Please, explain briefly.
Response 2: The soil used in this experiment was the humus soil purchased in the market (the nutrient soil formed by the decomposition and fermentation of plant branches and leaves in the soil by microorganisms). These soils were sterilized at 121 °C for 2 h before use. Therefore, the soil used in this experiment is defined as sterile nutrient soil. Additional explanations in the article are as follows:
Kidney bean (Phaseolus vulgaris) seeds (variety: Jinshulu) were obtained from the Shengnong Seed Company in Xinji City, Hebei Province, China. The seeds were planted with sterile nutrient soil (nutrient soil of sterilized at 121°C) in an artificial climate room [25 ± 1°C, 65 ± 10% RH, and a 14:10 h (L:D) photoperiod].
Thank you again
Best regards.
All authors